# miRNAs as Regulators of the Early Local Response to Burn Injuries

**DOI:** 10.3390/ijms22179209

**Published:** 2021-08-26

**Authors:** Ines Foessl, Christoph Walter Haudum, Ivan Vidakovic, Ruth Prassl, Joakim Franz, Selma I. Mautner, Sonja Kainz, Elisabeth Hofmann, Barbara Obermayer-Pietsch, Thomas Birngruber, Petra Kotzbeck

**Affiliations:** 1Department of Internal Medicine, Division of Endocrinology and Diabetology, Medical University of Graz, 8036 Graz, Austria; christoph.haudum@medunigraz.at (C.W.H.); joakim.franz@medunigraz.at (J.F.); selma.mautner@medunigraz.at (S.I.M.); barbara.obermayer@medunigraz.at (B.O.-P.); petra.kotzbeck@medunigraz.at (P.K.); 2CBmed GmbH—Center for Biomarker Research in Medicine, 8010 Graz, Austria; 3Gottfried Schatz Research Center (for Cell Signaling, Metabolism and Aging), Division of Biophysics, Medical University of Graz, 8010 Graz, Austria; ivan.vidakovic@medunigraz.at (I.V.); ruth.prassl@medunigraz.at (R.P.); 4HEALTH—Institute for Biomedicine and Health Sciences, JOANNEUM RESEARCH Forschungsgesellschaft mbH, 8010 Graz, Austria; sonja.kainz@joanneum.at (S.K.); thomas.birngruber@joanneum.at (T.B.); 5Department of Surgery, Division of Plastic, Aesthetic and Reconstructive Surgery, Medical University of Graz, 8036 Graz, Austria; elisabeth.hofmann@joanneum.at; 6COREMED—Cooperative Centre for Regenerative Medicine, JOANNEUM RESEARCH Forschungsgesellschaft mbH, 8010 Graz, Austria

**Keywords:** burn, skin, miRNAs, biomarkers, gene regulation, extracellular vesicles

## Abstract

In burn injuries, risk factors and limitations to treatment success are difficult to assess clinically. However, local cellular responses are characterized by specific gene-expression patterns. MicroRNAs (miRNAs) are single-stranded, non-coding RNAs that regulate mRNA expression on a posttranscriptional level. Secreted through exosome-like vesicles (ELV), miRNAs are intracellular signalers and epigenetic regulators. To date, their role in the regulation of the early burn response remains unclear. Here, we identified 43 miRNAs as potential regulators of the early burn response through the bioinformatics analysis of an existing dataset. We used an established human ex vivo skin model of a deep partial-thickness burn to characterize ELVs and miRNAs in dermal interstitial fluid (dISF). Moreover, we identified miR-497-5p as stably downregulated in tissue and dISF in the early phase after a burn injury. MiR-218-5p and miR-212-3p were downregulated in dISF, but not in tissue. Target genes of the miRNAs were mainly upregulated in tissue post-burn. The altered levels of miRNAs in dISF of thermally injured skin mark them as new biomarker candidates for burn injuries. To our knowledge, this is the first study to report miRNAs altered in the dISF in the early phase of deep partial-thickness burns.

## 1. Introduction

Burns are complex injuries with a multitude of local and systemic changes that are aggravated in severe and large lesions caused by full-thickness burns. Several local responses have already been described to be triggered by a severe burn injury [1,2,3], but the determinants of why and when a burn injury is causing the transition from a local to a systemic response are still widely unknown [4]. As a local response, immediately after the burn, the transcription activator nuclear factor-κB (NF-κB) is activated to regulate further inflammatory mediators. In the following pro-inflammatory phase [3], macrophages release mediators, such as interleukin-6 (IL-6), tumor necrosis factor alpha (TNF-α), prostaglandins and reactive nitrogen species [1]. Pro-inflammatory cytokines are released and accompanied by the formation of reactive oxygen species (ROS), and increased apoptosis is, in part, triggered by TNF-α [2]. The systemic responses to severe burns affect almost every organ system and are referred to as hypermetabolic response. The exact cause of this effect is still unclear, but an increased and prolonged expression of glucocorticoids, glucagon, catecholamines and dopamine, together with increased levels of cytokines, ROS, nitric oxide and several other mediators after a burn injury, lead to the hypermetabolic state in the patient that can last for years [5]. The first 48 h of the systemic response to a severe burn are characterized by a decrease in metabolism, cardiac output and oxygen consumption [6]. Surgical treatment of burn injuries in the early phase within the first 24 h after the burn has shown to improve patients’ prognosis in terms of inflammatory and hypermetabolic response [7]. However, the mere clinical assessment regarding depth and severity of such wounds is often imprecise and underestimated [8] and does not allow a prediction of the systemic response. Biomarkers for an evidence-based decision as to whether or not a burn wound should be surgically restored are needed to improve patient outcome. Therefore, an in-depth understanding of how local responses transition into systemic changes is highly relevant.

MicroRNAs (miRNAs) have been reported as reliable biomarkers for multiple disease types, including several skin conditions [9]. Differential expressions of certain miRNAs are described for psoriasis [10,11,12], atopic dermatitis [13,14] and keloid tissue [15]. In the denatured dermis of patients with full-thickness burns, 66 miRNAs showed differential expression four days after the burn, when compared to unburned skin [16], but to date, no biomarker has been established among these in clinical practice. Most of the biomarker studies for skin conditions have used plasma or serum samples [9,17,18], but local miRNAs might be masked by miRNAs from tissues that are in close contact with the bloodstream, such as blood cells, liver, kidney or lung tissue [19]. Locally, miRNAs have already been found in the dermal interstitial fluid (dISF) of rats and humans [20]. Because of the use of minimally invasive sampling techniques, such as dermal open-flow micro-perfusion (dOFM) [21], dISF is now more easily accessible [19] and offers the opportunity of measuring miRNA directly at the burned skin site. Reportedly, miRNAs are transported by extracellular vesicles [18], which have also been detected in dISF, where they were shown to mediate crosstalk of keratinocytes and fibroblasts in the context of aging [22]. Little is known about alterations in dermal extracellular vesicles/exosome-like vesicles (ELVs) at the burn injury site, but ELVs derived from human mesenchymal stem cells from the umbilical-chord were shown to accelerate wound healing in the burn wounds of rats [23].

The aim of this study was to identify miRNAs in the dISF that can be used as biomarkers for burn injuries and to assess their interaction with genes of the early burn response. We used a bioinformatics approach [24] to preselect potential miRNA–mRNA interaction partners by analyzing a publicly available microarray dataset [25] for changes in gene expression in the first 3 days after a partial-thickness burn. We used an established ex vivo human skin model [26] to study alterations within 24 h after a deep partial-thickness burn injury. We analyzed ELVs in dISF collected by using dOFM and followed miRNA tracks into the extracellular space. By using real-time PCR (qPCR), we screened burned skin tissue for a subset of both miRNAs and mRNAs.

## 2. Results

The study concept and workflow is depicted in Figure 1.

### 2.1. In the Early Response to Burn Injuries Putatively Involved miRNAs Are Identified through a Bioinformatics Approach

To identify regulated miRNAs after burn injuries, we set up a bioinformatics approach and analyzed publicly available GEO datasets from transcriptomic studies of biopsies collected from burn patients. To do so, the NCBI-GEO database [27] was screened for expression data in the early phase of partial-thickness burns of human skin biopsies. The GEO-dataset GSE8056 [25] was considered the most suitable at the time of the analysis. It included microarray data of 12 samples, three per group, each consisting of the pooled RNA of five patients. We compared three sample-pools from 0 to 3 days after the burn injury, with the three corresponding control sample-pools. Of the 54,675 annotated genes in the analyzed dataset, 3558 (6.5%) were differentially regulated with an adjusted *p*-value (p_adj_) of < 0.05, 992 (1.8%) with p_adj_ < 0.01 and 114 (0.2%) with p_adj_ < 0.001 (Table 1). These highly significant early burn-response genes (differentially regulated genes with p_adj_ < 0.001) (Table 2) were depicted in a heatmap (Figure 2) and considered for further analysis steps. Among these genes of the early burn response were matrix-associated genes, such as matrix metalloproteases (MMP) and collagens (COL); immunomodulatory genes, such as members of the interleukin (IL) and chemokine (c-x-c motif) ligand (CXCL) families; signal cascade transducers that activate various major cellular pathways; and several others. Of the 114 genes, 79 were upregulated in burn samples (70%) and 35 were downregulated (30%) (Table 1). This is also visualized in the volcano plot (Appendix A) and Venn diagram (Appendix A).

Within these 114 early burn response genes, miRNA target sites in their 3′UTRs were identified by using two platforms:

In TargetScanHuman, mRNAs of the early burn response genes were entered and conserved sites for miRNA families broadly conserved among vertebrates were recorded. In parallel, potential miRNA interaction partners for the early burn response genes were also scanned with miRWalk. We identified 113 miRNAs by TargetScanHuman and 251 miRNAs with miRWalk that potentially regulate the 114 early burn response genes. For further analysis, we selected those miRNAs that showed ≥5 interaction partners and/or were found as potential targets by both methods (Appendix A Appendix A). Thereby, 43 miRNAs were identified as potential regulators of early burn response genes. For 37 of these miRNAs, gene expression assays for analysis were commercially available (Appendix A Appendix A).

### 2.2. Extracellular Vesicles and Their miRNA Cargo in dISF of Burned Skin Have Been Characterized

To investigate the presence and characteristics of ELVs in the dISF of burn injuries, dOFM was performed as previously described [21,28,29], and ELVs were enriched through centrifugation (Figure 3). To verify the presence of ELVs, the exosomal markers flotillin and CD9 were detected in ELV-enriched fractions and mixed particle fractions (MPF) of burned skin and control skin (Figure 4a). There were no changes in the CD9 signal in the ELV-enriched fraction. A slight decrease in CD9 was present in the MPF of burned skin. Protein concentrations in the MPF were 0.8 mg/mL in burn samples and 0.5 mg/mL in control samples (*p* = 0.06). There were no changes in protein concentrations in the ELV-enriched fractions of burn sites (Figure 4b). Particle sizes in ELV-enriched fractions were not altered in the dISF of burned skin (Figure 4c and Appendix A).

### 2.3. After a Burn Injury miR-497-5p Is Downregulated in dISF

Of the 37 miRNAs analyzed as potential regulators of early burn response, 32 were detected by qPCR in the ELV-enriched fractions of dISF, collected within the first 4 h after a burn (Appendix A Appendix A). Five miRNAs could not be detected, namely miR-182-5p, miR-183-5p, miR-302c-3p, miR-367-3p and miR-372-3p. Three miRNAs were altered in ELV-enriched fractions. In tissue and in dISF of burn sites, miRNA-497-5p was downregulated. For miR-218-5p and miR-212-3p, a downregulation was observed in dISF, but not in tissue (Figure 4d and Figure 5).

We created a potential interaction network (Appendix A) of miRNAs in dISF (black and red) with the selected mRNA target genes (blue). The miRNAs found to be differentially regulated in dISF are displayed in red and highlighted by their name next to the interaction point. The miRNAs that showed differential expression were tightly embedded within the network and they were potential regulators of several genes that are differentially regulated in early burn response. For miR-497-5p, five interaction points were recorded; miR-218-5p showed six network interactions; miR-21-5p interacted with four partners; and miR-212-5p had one interaction.

### 2.4. A Subset of miRNAs and Genes Is Differentially Expressed within the First 24 h in Burned Tissue

The early burn-response genes that we identified through bioinformatics were only available from a mixed pool from biopsies derived from burn injuries from 1 up to 3 days post-burn. We were interested in identifying genes that were deregulated in the first 24 h after burn injury. To study the regulation within the first 24 h after a burn injury, we analyzed a new set of skin biopsies by using an established ex vivo human skin model. Healthy human skin explants from abdominoplastic surgeries were inflicted with a contact burn injury by using a heated steel block [26]. At time points 1, 4 and 24 h after burn injury, skin biopsies were collected, and expression of selected miRNAs (Table 3 and Appendix A Appendix A) and mRNAs (Table 3 and Appendix A Appendix A) were compared to control samples taken at the same time points.

In tissue, we analyzed 11 miRNAs out of the 43 miRNAs identified. We analyzed miR-7-5p, miR-16-5p, miR-21-5p, miR-101-3p, miR-182-5p, miR-195-5p, miR-212-3p, miR-218-5p, miR-221-3p, miR-424-5p and miR-497-5p in tissue biopsies (see Figure 5 and Table 3). We found that all 11 analyzed miRNAs were expressed in skin biopsies, with three of them differentially expressed in a time-dependent manner (Figure 5). Of the 11 analyzed early burn response genes, eight showed differential expression at least at one time point (Figure 6). DEAD-box helicase 3 Y-linked (DDX3Y), peptidase inhibitor 15 (PI15) and endothelial lipase (LIPG) showed upregulation in burned tissue after 1 h. TIMP Metallopeptidase Inhibitor 3 (TIMP3), Dickkopf-related protein 2 (DKK2), membrane metalloendopeptidase (MME), tenascin C (TNC) and solute carrier family 2 member 3 (SLC2A3) showed an increase in expression at 24 h. There were no differences observed for the expressions of WNT2B, a member of the wingless-type MMTV integration site (WNT) family, SRY-Box transcription factor 5 (SOX5) and metallothionein-1G (MT1G) within the first 24 h post-burn (Figure 6).

### 2.5. In Burned Skin Tissue miRNAs and Their Potential Targets Are Regulated in a Time-Dependent Manner

In tissue, miR-101-3p was upregulated in the early phase 1 h after the burn stimulus, and its expression declined to control levels at later time points. DDX3Y is one of the genes putatively interacting with miR-101-3p (Table 3). DDX3Y was upregulated in skin biopsies of burn injuries at 1, 4 and 24 h post-burn (Figure 6). The expression of miR-497-5p, another interaction partner of DDX3Y, was decreasing over time post-burn. Downregulation 4 h after the burn was shown for miR-221-3p, and this was followed by an upregulation of its interaction partners TIMP3 and DKK2 at 24 h after the burn injury. MiR-497-5p was downregulated in tissue 4 h post-burn. Its interaction partners were WNT2B, SOX5, MT1G, DDX3Y and SLC2A3. Among them, DDX3Y was upregulated at all time points, and SLC2A3 was upregulated 24 h post-burn.

## 3. Discussion

In this study, we analyzed miRNAs and gene expression of the early burn response in human skin tissue after a deep partial-thickness burn. Gaining insight into local tissue alterations in the early phase of skin burns could identify biomarkers for evidence-based treatment options. To overcome the struggle of clinical sample collection in vivo, we used an established ex vivo human skin model for burn injuries. This allowed us to study the early phase after a burn in a time-dependent manner. In order to screen for biomarkers in the skin without having to take invasive tissue samples, we focused on dISF, a fluid that is accessible with the minimally invasive dOFM method. We chose dISF instead of other body fluids, such as plasma, due to the close contact of dISF with the skin cells which enables a better representation of local conditions.

In our model, miR-497-5p was continuously downregulated in the tissue 1, 4 and 24 h post-burn, and this downregulation was reflected in dISF. Of note, miR-497-5p has been reported as a regulator of fibroblast viability. Recent work showed that miR-497-5p was upregulated in hypertrophic scars [30], which often occur after burn injuries [31]. In malignant melanoma tissues, miR-497-5p was found significantly downregulated [32]. Various cancer studies reported miR-497-5p as a tumor regulator [32,33,34], potentially through its regulatory effects on NF-κB [35] and FGF-2 signaling [36]. Hofmann et. al. were able to show that FGF2 was upregulated to a similar extent, while miR-497-5p was downregulated in our study [26]. Effective downregulation of miR-497-5p may therefore be an indicator for the regeneration capacity of a burn injury.

We found miR-21-5p downregulated in dISF of partial-thickness burns, but not in skin biopsies. A potential source of miR-21 might be mesenchymal stem cells, since miR-21 is known to regulate adipogenic differentiation through TGF-β signaling [37]. In healthy skin, miR-21 was found highly expressed in adipocytes and sebaceous glands, but not in the epidermis of mice. In the same study, during wound healing of full-thickness skin excisions, miR-21 was found highly expressed in the epidermis around the wound, especially at the sites where epithelial cells were migrating. In granulation tissue, miR-21 was also found upregulated in mesenchymal cells. Furthermore, miR-21-antagomir was shown to inhibit wound healing in mice by inhibiting collagen deposition [38]. MiR-21 is also reportedly upregulated in many cancer types [39,40]. This might indicate a contribution of miR-21-5p to the wound healing of burns. The skin cells might take up part of the surrounding miR-21-3p to support cell growth and migration.

We found miR-218-5p downregulated in dISF of burn injuries. MiR-218-5p was reported to enhance apoptosis by targeting secreted frizzled-related protein 2 (SFRP2), which, in turn, is an activator of the Wnt signaling pathway [41]. We did not observe alterations in WNT2B expression in the skin tissue within the first 24 h post-burn in our study. MiR-218 increased apoptotic cell death in the context of lung cancer [42]. Local downregulation of miR-218-5p could decrease apoptosis in the skin, thereby acting as a protection mechanism against cell death.

In our study of deep partial-thickness burns, we found miR-221-3p downregulated in tissue 4 h post-burn. This resembles the findings of Liang et al. in denatured dermal tissue of deep burn patients, where a profile of 66 miRNAs was found altered, including a downregulation of miR-221 [16] at day four post-burn. MiR-221-3p is reportedly downregulated in human skin fibroblasts exposed to bacterial lipopolysaccharides (LPS), a model for infected skin wounds, and overexpression of miR-221-3p reduced the negative effect of LPS on cell growth [43]. In the serum of psoriatic patients, miR-221-3p was upregulated, and lower expression inhibited cell growth in culture [44]. Therefore, miR-221-3p might be a regulator of fibroblast regeneration. This is supported by the fact that miR-221 was found downregulated in keloid skin tissue [15]. The control of miR-221-3p levels may be beneficial for the recovery of skin from inflammatory processes. Our data show that miR-221-3p was downregulated 4 h post-burn, and the levels were restored to control levels at 24 h, which is controversial to the findings of Liang et al. which showed its downregulation in burned skin after 4 days in denatured dermis of deep burn wounds. We thus speculate that miR-221-3p might be a candidate marker for the severity of burns. To clarify that, further experiments are needed to elucidate the expression of miR-221-3p over time in different degrees of burn wounds.

In another study, one in which human dermal fibroblasts were heat-shocked and their altered miRNA-expression was determined, neither the expression of miR-221-3p nor miR-497-5p was altered [45], indicating that other cells in human skin might be the source of these two miRNAs. Extracellular miR-497-3p could derive from Langerhans-cells, dendritic cells, or T-cells in the skin [46] as paracrine signal to the tissue.

In the tissue, 1 h after the burn stimulus, miR-101-3p was upregulated, and its expression declined to control levels at later time points. MiR-101-3p might thus be an early response regulator of skin burns, potentially by fine-tuning the upregulation of its interaction partner DDX3Y in the first hour after a burn.

Little is known about miRNA expression in the early phase after a partial-thickness burn injury. Guided by a bioinformatics approach, we identified potential miRNA–mRNA interaction partners of the early burn response. We are aware that this biased approach holds its limitations. However, we chose it to identify miRNAs that might specifically target genes of the early burn response, allowing for the exclusion of genetic noise that is a great struggle with unbiased methods, such as miRNA sequencing [47]. A combination of target prediction tools in order to minimize false-positive findings by in silico approaches has previously been described [48].

In line with the observation of a general upregulation of genes in burned skin in the here-analyzed GEO dataset and in the literature [49], we found most miRNAs in our analysis downregulated. Whether this downregulation of miRNAs is an active regulatory mechanism or a secondary effect needs further investigation. In our study, TIMP3, DKK2, MME, TNC and SLC2A3 showed an increase in expression only at 24 h, indicating that there is a time dependency in the onset of the upregulation of certain genes. That might also explain why we could not find differences in the expressions of WNT2B, SOX5 and MT1G while there was a difference in the GEO-dataset analyzed. Since the timeframes of sampling are different for the two analyses, namely 3 days for the GEO-dataset and 24 h for the skin burn model, these latter genes might only be upregulated at time points later than 1 day.

MiRNAs can act as intercellular messengers [50,51] and are packed in ELVs for that purpose [52]. Those ELVs are then released into the extracellular fluid [18]. Therefore, we claim that dISF is the ideal fluid to examine such intercellular communication via miRNAs, in the context of signaling from skin injuries. Motivated by the growing interest of the scientific community in minimally invasive sampling, we analyzed the feasibility of dOFM as a tool for the sampling of miRNAs and their further analysis. MiRNAs in extracellular body fluids are found to be surprisingly well protected from RNase degradation. They are packed into exosomes and protected by carrier proteins [34]. Exosomes and their miRNA cargo have been reported to impair wound healing and epidermal differentiation in mouse models [53]. Furthermore, they are mediators in the cellular crosstalk between fibroblasts and keratinocytes in skin aging [22]. However, little is known about their function in the early burn response. As carriers of miRNAs, they might play pivotal roles in the local cellular response to a burn stimulus and could serve as potential biomarkers.

We were able to show the presence of ELVs in dISF of burned skin. The dilution of the dISF as a result of the sampling method led to low concentrations in the final samples; thus, isolation methods should be further improved. Both fractions with possible ELV content stained positive for the exosomal markers flotillin and CD9 and showed a characteristic size pattern, as determined by nanoparticle tracking analysis. We did not find any major differences in ELV size and count but we observed a tendency towards an increase in protein concentration in the MVP fraction post-burn. This might be caused by an increased secretion of proteins and ELVs from the heat-stressed tissue [54]. However, further research is needed to investigate the contribution of ELVs in detail in the context of burn injuries.

In this work, we showed the presence of miRNAs in dISF and suggest miRNA-497-5p as a potential biomarker for the early burn response. We are aware that this study holds general limitations. One limiting factor is the small sample size in the bioinformatics analysis, as well as the limited amount of ex vivo skin model samples. The stringent inclusion criteria of p_adj_ < 0.001 for the selection of the early burn response genes was chosen to reliably identify targeting miRNAs. Based on our data presented in this manuscript, a follow-up study with skin samples collected at days 0–3 post-burn should be conducted to sequence mRNA, as well as miRNA from tissue, dISF and serum miRNAs from the same patients in order to elucidate the feasibility of miR-497-5p and other miRNAs as biomarkers for the early burn response.

To conclude, this study has identified miRNAs that are potentially involved in the early burn-response gene regulation. We found miRNA-497-5p, a known regulator of skin cell regeneration, downregulated in tissue and dISF of burned skin. Therefore, we propose the further examination of miR-497-5p as a biomarker for the severity of burn wounds, potentially through signaling via ELVs.

## 4. Materials and Methods

### 4.1. Tissue Samples

Fresh abdominal human skin explants from adult Caucasian donors were obtained from 10 donors, with a mean age of 36 (range = 51–25; 8 females and 2 males) undergoing abdominoplasty or circumferential body-lift surgery at the Division of Plastic, Aesthetic and Reconstructive Surgery, Department of Surgery, Medical University of Graz, Austria. The subcutaneous lipid layer and skin integrity of the skin explants were kept intact to ensure maximum resemblance to the in vivo situation.

Study approval was given by the Ethical Committee of the Medical University of Graz, Austria (EK: 28-151 ex 15/16; approval extended until 22 December 2020). All subjects gave written informed consent. The tissue transport and pseudonymization was carried out by Biobank Graz (Medical University of Graz, Austria). The study was carried out in accordance with the principles of the Declaration of Helsinki.

### 4.2. Human Skin-Burn Injury Ex Vivo Model and Dermal Open Flow Microperfusion (dOFM)

For a detailed description of the previously described method, please refer to References [26,55]. Shortly, fresh skin explants were cleaned by using gauze and water and immobilized with cannulas on a plastic-wrapped polystyrene plate. Then dOFM implantation sites were marked by using a stencil and a permanent marker. After that, dOFM probes were inserted with the help of guide cannulas. To monitor the dermal temperature, a temperature sensor was inserted and connected to a temperature data logger. Implantation sites were sealed with cyanoacrylate adhesive. The dOFM probes were connected to the tubing, the filled perfusate bags and the peristaltic pumps. Between the outlet of the dOFM probe and the pull tubing, a sampling tube was connected. At the run-in phase, dOFM probes were flushed with a flow rate of 10 µL/min for a maximum of 5 min. Afterwards, the flow rate was reduced to 1 µL/min. After 30 min, the run-in samples were discarded, and the regular sampling tubes were attached. For the burn stimulus, a preheated (100 °C) stainless-steel block of 5 × 5 cm, 1.9 kg (Zultner Metall GmbH, Graz, Austria), was placed on the skin for 10 s, without additional pressure. Sampling was performed for 4 h. At 1, 4 and 24 h, biopsies were taken from the burned sites and from the control areas. Biopsy material was snap-frozen on dry ice and stored at −80 °C until further processing.

### 4.3. Gene-Expression Data Analysis and Selection of miRNA-Targets

Data freely available on the GEO gene expression omnibus database [27] were screened for miRNA expression in the early phase, preferably within the first days, of deep partial-thickness skin burns. Since such miRNA expression data were absent, our search was extended to RNA expression. The most suitable dataset found was “Gene Expression Profiles in Thermally Injured Human Skin: A Temporal Microarray Analysis” with the accession number GSE8056 from Greco et al. (2010) [25], who analyzed it in silico. From the total of 12 samples in the microarray dataset, the 3 datasets “normal human skin, pooled replicate #1–3 of normal (no thermal injury) skin control” with accessions GSM198875, GSP198876 and GSM198877 were selected as controls. These were compared to the samples “burn wound margin of human skin, replicate #1–3 of pooled time group 0–3 days post-thermal injury” with accessions GSM198866, GSM198867 and GSM198868. Differential expressions between control and burn samples were determined with GEO2R with log-transformed data, calculating false discovery rate (Benjamini–Hochberg adjusted *p*-values, p_adj_). Significance level cutoff was set at 0.001. Visualization of the GEO2R analysis is depicted in Appendix A. The genes with an adjusted *p*-value of <0.001 were analyzed with TargetScanHuman 7.2 [24] for conserved sites for miRNA families broadly conserved among vertebrates and with miRWalk 2.0 [56]. The interactive interaction network (Appendix A Appendix A) was created with RStudio (2020, Integrated Development for R. RStudio, PBC, Boston, MA, USA) and the networkD3 package.

### 4.4. RNA Extraction and qPCR

Homogenization of skin biopsies was performed in Qiazol, using MagNa Lyser Beads and the MagNA Lyser instrument (Roche, Basel, Switzerland). RNA was isolated with RNAeasy Lipid Tissue Mini Kit (Qiagen, Hilden, Germany) immediately after homogenization, according to the manufacturer’s instructions. RNA concentration was determined on a NanoDrop microvolume spectrophotometer (Thermo Fisher Scientific, MA, USA). For cDNA synthesis (iScript gDNA clear kit, Biorad, CA, USA) 0.5–1 μg of total RNA was used. Predesigned TaqMan assays for the genes of interest and the endogenous control genes (Appendix A Appendix A) and the TaqMan Gene Expression Master Mix were purchased from Thermo Fisher Scientific, MA, USA. For miRNA expression from tissue, RNA was extracted with RNeasy Mini Kit including Spike-In controls, and cDNA synthesis was performed with miRCURY LNA RT Kit and miRCURY LNA SYBR Green PCR Kit was used with predesigned miRCURY LNA miRNA PCR Assays (Appendix A Appendix A) (all from Qiagen, Hilden, Germany).

MiRNA extraction from ELV fractions was performed with an miRNeasy Serum/Plasma Advanced Kit, according to manufacturer’s protocol, including Spike-In controls (UniSp2, UniSp4 and UniSp5), followed by cDNA synthesis with miRCURY LNA RT Kit (including Spike-Ins UniSp6 and cel-miR-39-5p). For qPCR, the miRCURY LNA SYBR Green PCR Kit was used with miRCURY LNA miRNA PCR Assays (all from Qiagen, Hilden, Germany) (Appendix A Appendix A), according to the manufacturer’s instructions.

### 4.5. qPCR Analysis

Relative gene expression was calculated by using the minus delta–delta Ct (-ddCt) method [57]. For gene expression in tissue, target gene expression was normalized to the averaged Cq of the three endogenous control genes (TPB, RPLP0 and GAPDH). For miRNA expression in tissue, the averages of SNORD48 and U6 were used for normalization. For miRNAs in dISF, samples were calibrated with an interplate calibrator. Spike-ins were used to determine the quality and homogeneity of RNA extraction and cDNA preparation.

Expression levels were determined as duplicates from samples of at least three independent experiments. Individual values were calibrated to the average of the control samples and are presented as individual -ddCt values with mean (line) and standard deviation (whiskers). All qPCR experiments were run on a CFX384 cycler (Bio-Rad, Hercules, CA, USA), using standard conditions, according to the manufacturer’s instructions.

### 4.6. Sample Processing and ELV Enrichment

After 4 h of dOFM sampling from ex vivo skin explants, perfusates were stored at 4 °C and processed immediately. A perfusate volume of approximately 60 µL/probe/hour was collected. ISF was pooled from 2 adjacent probes in order to get a sufficient sample amount for further processing. Occasional blood smears were not aspirated, and hemolytic samples were carefully recorded by visual assessment. To remove debris, ISF was centrifuged at 500× *g* for 30 min. All centrifugation steps were performed at 4 °C. To remove apoptotic bodies (ApoBD) and cell fragments, supernatant was transferred to a fresh tube and centrifuged at 2000× *g* for 20 min. To pellet microvesicles, the supernatant was transferred to a fresh tube. Then 50 µL was removed as mixed particle fraction (MPF), and the remaining supernatant was centrifuged at 20,000× *g* for 70 min. The non-visible pellets from each centrifugation step (except the debris) were resuspended in 50 µL sterile-filtered PBS and stored at 4 °C overnight for nanoparticle tracking. For RNA extraction, the samples were frozen at −20 °C for short-term storage.

### 4.7. Nanoparticle Tracking Analysis

For nanoparticle tracking, dISF samples collected by OFM were stored at 4 °C overnight and measured the next day. MVP and ELV fractions were diluted with filtered deionized water (Whatman Anotop^®^ 25 Plus syringe filter, pore size 20 nm) 1:10 or 1:20 (v/v), depending on particle concentration. Samples were measured with the NanoSight LM10 (Malvern Panalytical Ltd., Kassel, Germany) with a 532 nm (green) laser at 25 °C, with the following settings: camera level 14, slide shutter at 1239, slider gain 366, number of frames 1499 and script SOP standard measurement. Five independent measurements were taken and analyzed for average particle properties in terms of size and numbers.

### 4.8. Protein Concentration and Immunoblot Analysis

Protein concentration was determined with Bio-Rad Protein Assay Dye Reagent (Bio-Rad Laboratories GmbH, Vienna, Austria), according to manufacturer’s protocol. Then 13.5 µg of protein was separated by SDS–PAGE and transferred to a PVDF membrane (Bio-Rad Laboratories GmbH, Vienna, Austria). Protein detection was performed with monoclonal antibodies for CD9 ((D8O1A) Rabbit mAb #13174) and flotillin-1 ((D2V7J) XP Rabbit mAb #18634) (Cell Signalling Technology, Boston, MA, USA). Specific antibodies were detected by an HRP-conjugated goat anti-rabbit antibody (Anti-rabbit IgG, HRP-linked Antibody #7074, Cell Signalling Technology, Boston, MA, USA). For loading control, Coomassie Blue staining of the membrane was performed. Bands were visualized by ChemiDoc™system with Clarity™ substrate (Bio-Rad Laboratories GmbH, Vienna, Austria).

### 4.9. Statistical Analysis

Statistical analysis was performed by using the software GraphPad Prism (version 8.0, San Diego, CA, USA). Data are presented as mean +/− SD. Two-sided Student’s *t*-test, followed by Bonferroni correction, was used to compare the data. A *p* < 0.05 was considered statistically significant, with *, ** and *** indicating *p* < 0.05, *p* < 0.01 and *p* < 0.001 in the graphs.

## Figures and Tables

**Figure 1 ijms-22-09209-f001:**
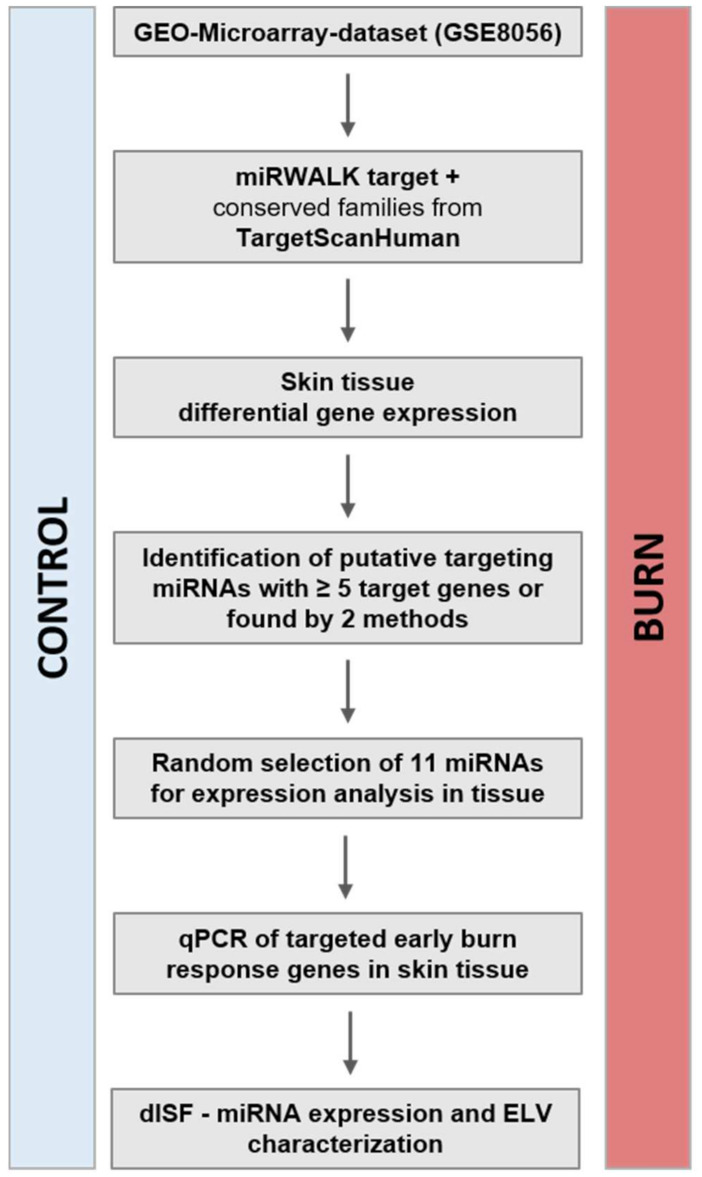
Study concept and workflow. From the publicly available dataset of GSE8056, we compared the groups for control (no burn) and the sample pools of up to 3 days after the burn wound. The genes with a p_adj_ < 0.001 were considered the genes of the early burn response. The miRNAs putatively targeting these mRNAs were identified via the bioinformatics approach (miRWalk target, TargetScanHuman). Selected miRNAs and the early burn response genes they target were analyzed in a human skin model for burn injuries. The miRNAs in human dISF were characterized. Abbreviations: GEO, Gene Expression Omnibus; miRNA, microRNA; qPCR, real-time quantitative PCR; p_adj_, adjusted *p*-value; dISF, dermal interstitial fluid; ELV, exosome-like vesicles.

**Figure 2 ijms-22-09209-f002:**
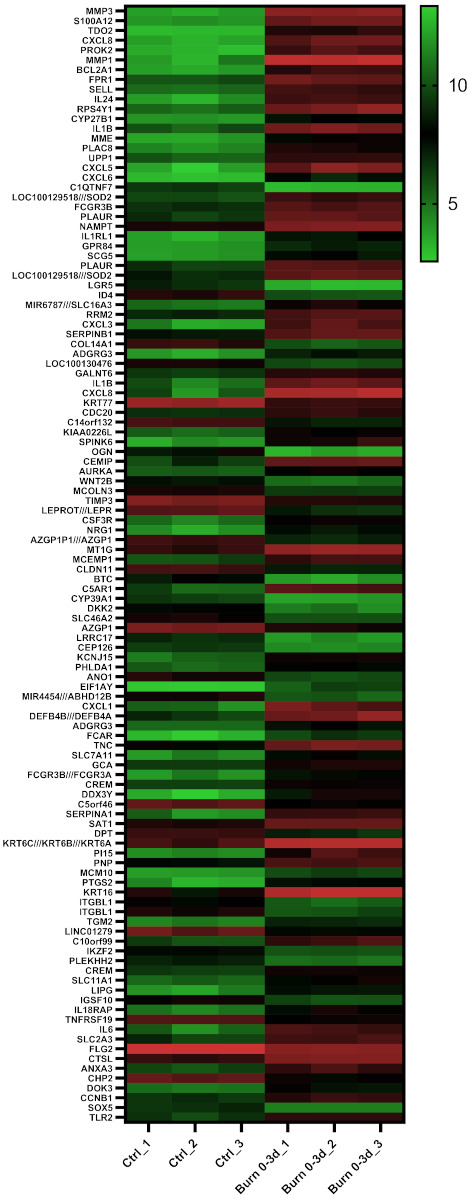
Heatmap displaying 114 early burn response genes that are differentially expressed in the GSE8056 dataset with p_adj_ < 0.001. The color indicates the expression, with green being downregulated and red upregulated. Data are displayed as log2 of normalized counts.

**Figure 3 ijms-22-09209-f003:**
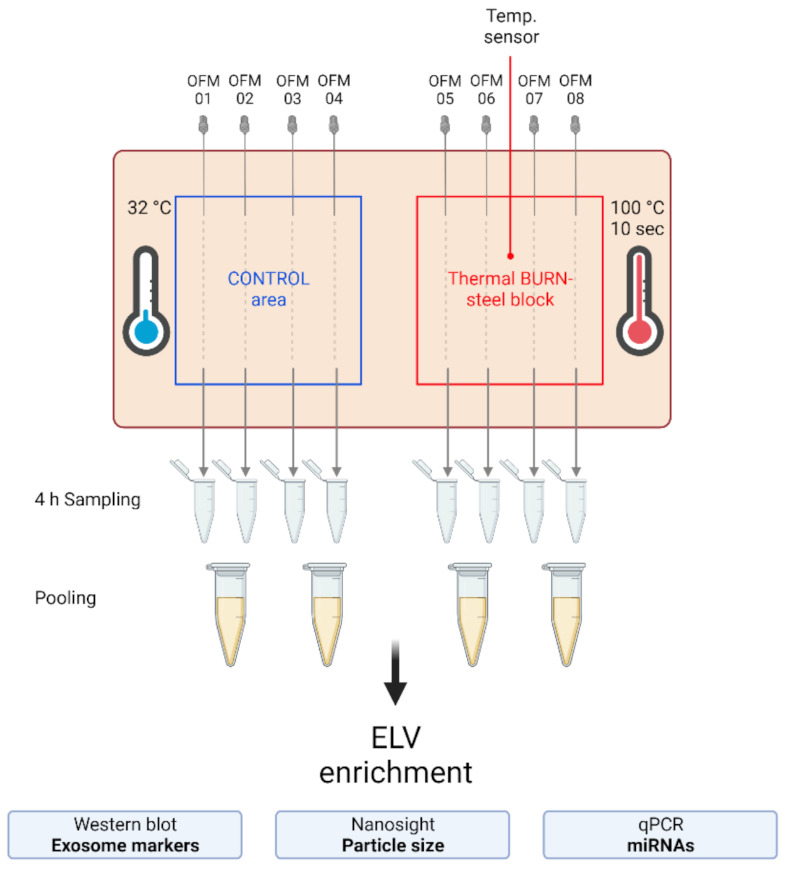
Experimental setup of the dISF sampling with dOFM in an ex vivo human skin model for burn injuries. OFM probes were inserted into ex vivo human skin at the control area (OFM 1–4) and burn area (OFM 5–8), where a steel block of 100 °C was applied for 10 s to provoke a thermal burn. A temperature sensor inserted into the skin enabled the control of temperature at the burn area. OFM was sampled over 4 h. Samples from 2 adjacent probes were pooled. The collected dISF was centrifuged at increasing speeds to enrich ELVs for further experiments.

**Figure 4 ijms-22-09209-f004:**
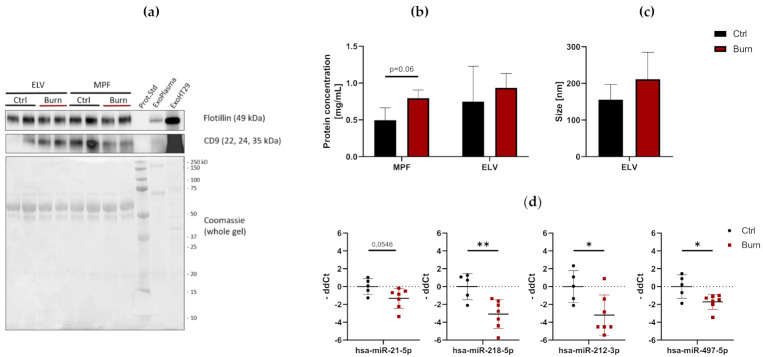
Characterization of dISF in an ex vivo human skin model for burn injuries. (**a**) Western blot for expression of exosomal markers flotillin and CD9. Coomassie staining of the whole gel is displayed as loading control. (**b**) Protein concentration in MPF and ELV and (**c**) mean size (nm) of particles in the ELV. (**d**) MiRNAs differentially expressed in dISF sampled with dOFM. Data are derived from 3 independent experiments and presented as individual values of -ddCt, normalized to an interplate calibrator and expressed relative to controls, with means (line) and standard deviation (whiskers). Significance was tested with Student´s *t*-test, followed by Bonferroni correction; *p*-values <0.05 were considered as statistically significant, with *, ** indicating *p* < 0.05, *p* < 0.01.

**Figure 5 ijms-22-09209-f005:**
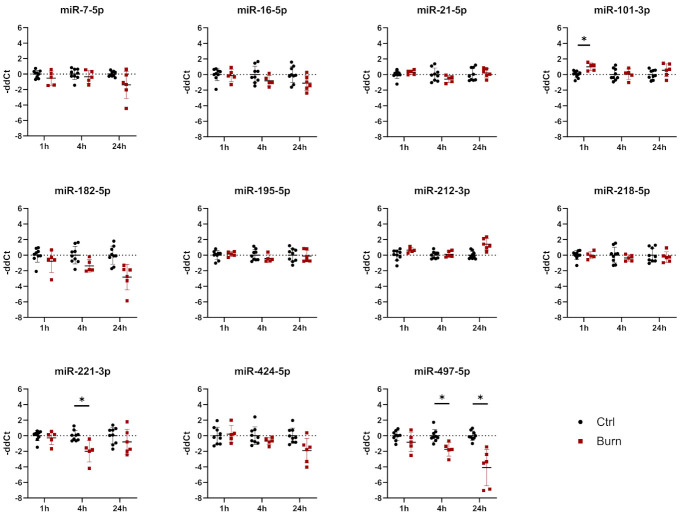
Expression of 11 selected miRNAs in skin biopsies putatively targeting early burn-response genes. Data are derived from 3 independent experiments and presented as individual values of –ddCt, normalized to SNORD 48 and U6 and expressed relative to control, with means (line) and standard deviation (whiskers). Significance was tested with Student´s *t*-test, followed by Bonferroni correction; *p*-values < 0.05 were considered as statistically significant, with * indicating *p* < 0.05.

**Figure 6 ijms-22-09209-f006:**
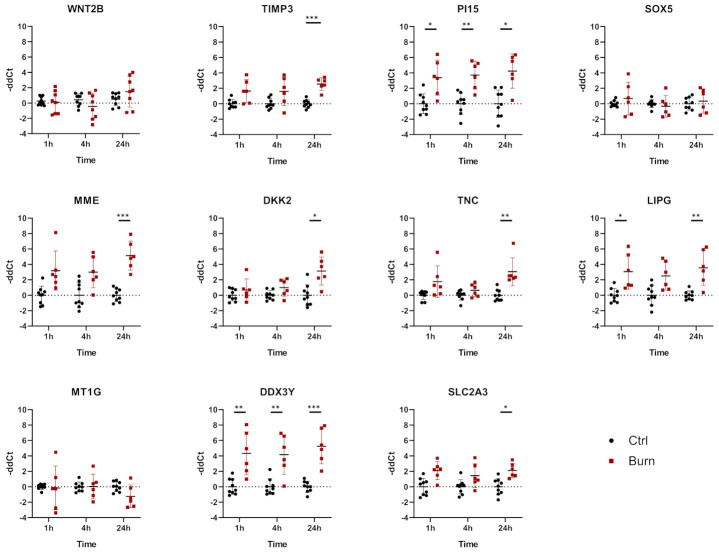
mRNA differential expression in biopsies from burned skin. Data are derived from 3 independent experiments and presented as individual values of –ddCt, normalized to TBP, RPLP0 and GAPDH and displayed relative to control samples, with means (line) and standard deviation (whiskers). Significance was tested with Student´s *t*-test, followed by Bonferroni correction; *p*-values < 0.05 were considered as statistically significant, with *, ** and *** indicating *p* < 0.05, *p* < 0.01 and *p* < 0.001.

**Table 1 ijms-22-09209-t001:** Results of the GEO2R analysis of the GSE8056 dataset. The groups for control (no burn) and the pooled burn samples 3 days after the burn wounds were compared.

	No.	%
Total genes annotated	54,675	100%
p_adj_ < 0.05	3558	6.5%
p_adj_ < 0.01	992	1.8%
p_adj_ < 0.001	114	0.2%
upregulated	79	70%
downregulated	35	30%

**Table 2 ijms-22-09209-t002:** Selected early burn-response genes, including their rank based on FDR (compare with heatmap Figure 2).

	ID	p_adj_	*P*	t	B	logFC	Gene Symbol	Gene Title	GSM 198875 0 h	GSM 198876 0 h	GSM 198877 0 h	Mean 0 h	GSM 198866 3 d	GSM 198867 3 d	GSM 198868 3 d	Mean 3 d
14	203434_s_at	0.0001246	3.19 × 10^−8^	19.65	9.10	4.28	*MME*	membrane metalloendopeptidase	12.02	11.80	16.98	13.60	211.84	278.65	300.48	263.66
49	238512_at	0.0006066	5.67 × 10^−7^	−13.73	6.78	−2.38	*WNT2B*	Wnt family member 2B	190.56	162.10	190.51	181.06	35.37	30.40	38.82	34.87
51	201150_s_at	0.0006066	5.88 × 10^−7^	−13.67	6.75	−2.35	*TIMP3*	TIMP metallopeptidase inhibitor 3	2684.61	2025.09	2336.93	2348.88	471.36	471.65	431.15	458.05
56	204745_x_at	0.0006066	6.22 × 10^−7^	13.57	6.70	2.62	*MT1G*	metallothionein 1G	642.73	474.11	677.58	598.14	3296.03	3958.74	3704.69	3653.15
62	219908_at	0.000613	7.14 × 10^−7^	−13.34	6.58	−3.11	*DKK2*	dickkopf WNT signaling pathway inhibitor 2	210.42	228.11	234.28	224.27	26.20	34.89	18.92	26.67
76	201645_at	0.000613	8.52 × 10^−7^	13.04	6.42	3.08	*TNC*	tenascin C	293.96	214.10	190.91	232.99	1506.81	2321.70	2087.41	1971.97
81	205000_at	0.0006873	1.02 × 10^−6^	12.75	6.26	4.96	*DDX3Y*	DEAD-box helicase 3. Y-linked	11.08	6.00	10.63	9.24	152.53	371.48	375.00	299.67
87	229947_at	0.0007308	1.16 × 10^−6^	12.54	6.14	5.05	*PI15*	peptidase inhibitor 15	18.37	22.90	19.28	20.18	332.87	1159.94	760.88	751.23
101	219181_at	0.0008774	1.62 × 10^−6^	12.02	5.83	3.30	*LIPG*	lipase G. endothelial type	16.70	12.60	27.13	18.81	196.16	170.26	164.54	176.99
106	202499_s_at	0.0009022	1.77 × 10^−6^	11.89	5.75	3.23	*SLC2A3*	solute carrier family 2 member 3	133.38	70.10	72.10	91.86	785.79	729.58	970.77	828.71
113	238009_at	0.0009914	2.07 × 10^−6^	−11.65	5.61	−2.13	*SOX5*	SRY-box 5	92.71	104.10	133.17	109.99	24.77	24.69	24.90	24.79

**Table 3 ijms-22-09209-t003:** Putative interactions between the analyzed miRNAs and mRNAs. The interactions are based on the TargetScan Human 7.2 search for conserved sites.

	WNT2B	TIMP3	PI15	SOX5	MME	DKK2	TNC	LIPG	MT1G	DDX3Y	SLC2A3
miR-7-5p	x										
miR-16-5p	x			x					x	x	x
miR-21-5p	x	x	x	x							
miR-101-3p										x	
miR-182-5p	x			x		x		x			
miR-195-5p	x			x					x	x	x
miR-212-3p				x							
miR-218-5p	x			x	x	x	x	x			
miR-221-3p		x				x					
miR-424-5p	x			x					x	x	x
miR-497-5p	x			x					x	x	x

## Data Availability

Data freely available on the GEO gene expression omnibus database [27] on “Gene Expression Profiles in Thermally Injured Human Skin: A Temporal Microarray Analysis” with the accession number GSE8056 from Greco et al. (2010) [25] were analyzed in silico.

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
