# Peer review of "miRNAs as Regulators of the Early Local Response to Burn Injuries"

_ijms, 2021, doi:10.3390/ijms22179209_

Round 1

Reviewer 1 Report

This is a well thought and well executed study, describing early molecular biomarkers of burn injuries in skin. Using in depth computational analysis approach, authors show a set of deregulated genes and their regulating miRNA partners in burn injuries. In addition this study uses ex vivo human skin model and qRT-PCR to further validate early changes in the expression of miRNA and their mRNA targets in skin burn injuries. The manuscript is well written and experiments are nicely conceived and performed, however there are following few points which needs to be addressed,    

  • In results section 2.1, this study uses GEO-dataset GSE8056 and authors described that this dataset was the most suitable for the analysis. It would more useful if the selection criteria for using this particular dataset is given.
  • Authors have used a very stringent selection criteria for differentially expressed genes in early burn response (table 1, padj<0.001). Although this analysis showed statistically robust changes in gene expression, the number of genes found by this analysis are very low and represents only 0.2% of the all the differentially expressed genes. This could potentially mask some of the key pathways involved in early response during burn injury. What are authors views on this?
  • Have authors thought about considering fold change and padj values together to get more number of differentially expressed genes. For example, padj<0.05/0.01 and more than +/- 2 to 5 fold change in expression. This could shed light on the genes regulating biological pathways important in burn injury.   
  • In figure 4d, out of 37 selected miRNAs we see changes in the expression of miR-21, miR-218, miR-212 and miR-497 in dISF. In the subsequent experiments authors have selected 11 miRNAs to be analyzed in tissue, what was the criteria to select other 7 miRNAs apart from the 4 miRNAs which showed differential expression in dISF?
  • miR-497 showed consistent downregulation upon burn injury both in dISF and tissue. Among its target only DDX3Y and SLC2A3 were upregulated in burned skin samples. However, DDX3Y was already upregulated at 1 hour a time point where miR-497 expression did not change, this suggest that upregulation of DDX3Y could be independent of miR-497. Also extent of DDX3Y expression did not change over time. Could authors check for miR-497 targets by using Targetscan and miRTarBase in GEO-dataset GSE8056 with less strict padj<0.01 values. This could potentially identify gene sets regulated by miR-497 and if they are enriched in burn injury and are responsible for regulating burn injury responses.
  • Is it possible to predict therapy response in burn injuries based on miR-497 expression? This would be a great addition to biomarker aspect of miR-497.
  • Some of the cited articles in the introduction section are old. Authors should cite recent articles wherever it is possible. For example, for differential expression of miRNAs in psoriasis, following articles are more recent and could be cited,

https://pubmed.ncbi.nlm.nih.gov/31207228/

https://www.medrxiv.org/content/10.1101/2021.01.27.21250590v1.full-text

  • There is some issue with the referencing tool as we see following error message at many places in the manuscript- (Error! Reference source not found.). This should be checked and sorted.

Author Response

Kind regards

Ines Foessl

Reviewer 2 Report

The paper "miRNAs as regulators of the early local response to burn injuries" written by Foessl et al presents identification of 43 miRNAs as potential regulators of the early burn response through the bioinformatics analysis of an existing dataset.

I think that the paper is of interest and worth to be published; I have only one minor point: The limitations of this paper are not really mentioned and discussed; please do so!

Author Response

Lind regards

Ines Foessl
